# Incidence and predictors of severe acute malnutrition mortality in children aged 6–59 months admitted at Pawe general hospital, Northwest Ethiopia

**Fassikaw Kebede**[1]*, **Tsehay Kebede**[2], **Belete Negese**[3], **Atitegeb Abera**[1], **Getahun Fentaw**[4], **Ayalew Kasaw**[5]

1 Department of Epidemiology and Biostatics, College of Health Science, Woldia University, Woldia, Ethiopia, 2 Faculty of Social Science, Department of Geography & Environment study, Bahir Dare University, Bahir Dar, Ethiopia, 3 Department of Nursing & Midwifery, College of Medicine & Health Science, Debre Birhan University, Debre Birhan, Ethiopia, 4 Department of Nutrition, School of Public Health, College of Health Science, Woldia University, Woldia, Ethiopia, 5 Department of Nursing and Midwifery, Gambela College of Health Science Southern Ethiopia, Gambela, Ethiopia

* fassikaw123@gmail.com

**Data Availability Statement:** All relevant data are included within the manuscript and Supporting information.

## Abstract

### Background

Severe acute malnutrition (SAM) is defined as a weight-for-height < -3z scores of the median WHO growth standards, or visible severe wasting or the presence of nutritional edema. SAM related mortality rates in under-five children are well documented in Ethiopia but data on their predictors are limited. We aimed to document factors associated with SAM related mortality to inform better inpatient management.

### Methods

A facility-based retrospective cohort study was conducted among children admitted due to SAM at Pawe General Hospital, Northwest Ethiopia, from the 1st of January 2015 to the 31st of December 2019. Data from the records of SAM children were extracted using a standardized checklist. Epi-Data version 3.2 was used for data entry, and Stata version 14 was used for analysis. Bi-variable and multivariable Cox regression analyses were conducted to identify predictors of mortality. Variables with P<0.05 were considered significant predictors of mortality.

### Results

Five-hundred sixty-eight SAM cases were identified of mean age was 27.4 (SD± 16.5) months. The crude death rate was 91/568 (16.02%) and the mean time to death was determined as 13 (±8) days. Independent risk factors for death were: (i) vomiting AHR = 5.1 (1.35–21.1, p = 0.026), (ii) diarrhea AHR = 2.79 (1.46–5.4, p = 0.002), (iii) needing nasogastric therapy AHR = 3.22 (1.65–6.26, p = 0.001), (iv) anemia AHR = 1.89 (1.15–3.2, p = 0.012), and (v) being readmitted with SAM AHR = 1.7 (1.12–2.8, p = 0.037).

**Funding:** The authors received no specific funding for this work.

**Competing interests:** The authors declare that there are no competing interests.

**Abbreviations:** AHR, adjusted hazard ratio; CHR, crude hazard ratio; CI, confidence interval; FMOH, Ethiopian Federal Ministry of Health; HU-CSH, Hawassa University Comprehensive Specialized Hospital; MUAC, mid-upper arm circumference; NGT, nasogastric intubation for feeding; PGRH, Pawe general and referral hospital; SAM, severe acute malnutrition; SC, stabilizing center; SD, standard deviation; WFH, weight for height.

## Conclusion

SAM mortality was high in under-five children in our setting. The identified risk factors should inform treatment and prevention strategies. Improved community health education should focus on healthy nutrition and seeking early treatment. Inpatient mortality may be reduced by stricter adherence to treatment guidelines and recognizing early the key risk factors for death.

## Introduction

Malnutrition remains one of the most common causes of morbidity and mortality in children throughout the world. It is responsible directly or indirectly for 60% of the 10.9 million deaths annually among under-five children and two-thirds of these deaths occur during the first year of life [1]. Childhood under-nutrition incorporates a combination of nutrition disorders that include underweight, wasting, stunting, and micronutrient deficiency [2,3]. Underweight (low weight-for-age) is a composite measure of wasting and stunting (low height-for-age) while wasting (low weight-for-height) is acute malnutrition due to a recent failure of nutrition (e.g. lack of food) or a recent infection like diarrhea causing weight loss. Severe acute malnutrition is defined as either a weight-for-height < -3z scores of the median WHO growth standard, a mid-upper-arm circumference (MUAC) < 115 mm, visible severe wasting, or the presence of nutritional edema [2,4]. Although SAM occurs globally and may affect all ages, infants and young children are most vulnerable, as they have higher nutritional requirements for growth and development [5], and sub Saharan Africa bears the greatest burden of SAM [6]. The peak age for SAM is 6–18 months, which is the time of fast growth and brain development [7].

Globally, in 2018 one in 12 of the estimated 52 million children under five had SAM [8], and 2.9 million of these children were admitted for inpatient treatment [8,9]. Despite the availability of outpatient treatment, 50% of SC admitted children with SAM die due to inappropriate care [10] and one important reason is poor adherence to SAM therapeutic guidelines [11]. Other factors include the presence of danger signs seen in sicker children like lethargy, hypoglycemia and hypothermia as well as bradycardia, capillary refill > 2 seconds, weak pulse volume and impaired level of consciousness [12].

The World Health Organization (WHO) has developed SAM management guidelines that, ifstrictly followed, should reduce the mortality to less than 10%. Mortality from SAM has, however, persistently remained between 10 and 40% in many hospitals in Sub Saharan Africa, despite the use of these guidelines. Malnutrition in Ethiopia has long been a contributing cause of death in infants and young children with estimates of 270,000 deaths each year [13]. According to the 2014 Health and Health-Related Indicators (HHRI), SAM was the third leading cause of mortality in Ethiopia and accounted for 8.1% of all deaths in under-five children [2,14,15]. This high mortality has been commonly attributed to HIV infection, lack of maternal participation in feeding programs, inadequate care and prescription errors, and over-prescription of intravenous therapies and blood transfusions. SAM in Ethiopia still accounts for 20% of pediatric hospital admissions and 30% of inpatient deaths [16]. Few studies have been conducted in Ethiopia to determine the predictors of inpatient mortality in SAM affected children. Indeed, no such study has been done in North West Ethiopia yet this region has the highest prevalence of SAM in Ethiopia. We, therefore, assessed the incidence and predictors of SAM related mortality rate in children under five undergoing inpatient treatment for SAM.

## Methods and materials

### Study area, design and setting

The study was conducted in the Pediatric ward of the Pawe general and referral hospital (PGRH) in the North Western province of Benishangul Gumuz, which is 560 km from Addis Ababa, the capital city of Ethiopia [17,18]. According to the 2019 national population projection, this region has an estimated 1.21 million inhabitants [19]. The Pediatric ward has 152 beds and a separate SC center for children with SAM. Ethiopia has adopted the WHO SAM treatment guidelines of under-five children using three phases: phase I, the transition phase, and phase II. In all phases, admitted children are treated empirically for infections, hypoglycemia, and resuscitated to restore electrolyte balance.

### Study design

This was a facility-based retrospective cohort study that was conducted among SAM children under five years who were admitted to PGRH, Northwest Ethiopia, from the 1st of January 2015 to the 31st of December 2019.

### Sample size determination

We determined the sample size using the single population proportion formula: n = (Za/2)2P (1-P)/d$^2$ and a 95% confidence level (Za/2) = 1.96. Assuming an overall mortality rate from research 46% [3], a margin of error of 5%, and allowing an additional 15% for incomplete data, the sample size was 438 children. In the event, we found 578 records between January 1, 2015 and December 31st 2019 and included them in the study.

### Outcome ascertainment

The dependent variable was death while an inpatient. The independent variables assessed included medical characteristics of children at enrolment.

### Definitions

Admission criteria: The WHO SAM definition for children aged 6–59 months of age was used; either a weight-for-height < -3z scores of the median WHO growth standards, or MUAC <115 mm, visible severe wasting, or the presence of nutritional edema [20]. Discharged/ declared cured: this was defined as a child whose weight-for-height/length is ≥ -2z scores without edema for at least 2 weeks or a MUAC >115 mm and no edema for at least 2 weeks [5].

Defaulted/lost to following up: a child who was not seen for at least two consecutive days after starting treatment [2]. Anemia in children defined as hemoglobin concentration <11 g/ dL [21].

### Data collection instrument and quality controls

A standard and pre-tested data extraction form was used to extract the required information from the case notes of SAM children [1]. Before the actual data collection, the prepared checklist of variables was pretested on 28 case notes of SAM children from Jawi primary hospital. A 2-day training was given for three diploma nurses and one BSc public health officer on the objectives of the study, variables of interest and maintaining data confidentiality. Strict followup and supervision were carried out during data collection by the principal investigators and feedback was given on a daily basis. The collected data were checked for inconsistencies, coding errors, completeness, accuracy, clarity, and missing values.

## Data processing and analysis

Data were entered using Epi-Data version 4.2 statistical software and exported to STATA (SE) R-14 version statistical software for further analysis. We used Cox proportional hazards regression models with robust sandwich covariance matrix estimates to account for repeated measurements for each child. Kaplan–Meier survival analysis was used to determine the cumulative probability of death for all children and the mean time to death. Variables with P-value < 0.25 in the bi-variable Cox regression analysis were included in the multivariable Cox regression model. We tested the assumptions of the Cox Proportional Hazards model using Schoenfeld residuals. Variables with an adjusted hazard ratio (AHR) and 95% confidence interval (CI) and a P value <0.05 were considered as significant predictors of inpatient mortality.

## Ethical statement and consent

The institutional review board (IRB) of Debre Markos University approved the study protocol Ref. No: HSC/984/16/12. A formal letter was submitted to PGRH requesting permission for the data collection. Data security and participants' confidentiality were maintained at all levels of data management.

## Results

### Baseline socio-demographic and clinical characteristics

After excluding 10 (1.74%) files due to incompleteness, we reviewed 568 files of SAM cases registered for treatment from 1st January 2015 to December 31 2020. Of the included children (Table 1), slightly more than half, 324 (57.04%) were females. The majority of children 356 (62.68%) were aged 6–24 months and the mean age was 26.28 (SD = ±16.04) months. Just over three quarters were rural residents, almost two thirds were breastfeeding, and the majority, 457 (80.46%), were newly admitted SAM cases. More than half (318) of the SAM cases were due to wasting and the remaining 153 and 97 had marasmus-kwashiorkor and kwashiorkor, respectively. Wasting was observed mostly in the 6–24 months age group.

### Co-morbidities

More than half the children were febrile on admission and 217 had dermatitis; 307 cases had comorbidities like SAM with diarrhea, pneumonia or anemia. 296 (52.11%) children had pneumonia at baseline, while 42 (7.39%) had a positive blood film for malaria. 528 (92.96%) children were HIV negative. Supplementary treatments included vitamin A, folic acid, deworming, and blood transfusions.

### Inpatient treatment outcomes

Of the 568 children, 91 (16.02%) died while in hospital with 32 and 24 deaths occurring in phase I and the transition phase, respectively (Table 2). The mean time to death was 13 (±8) days. Of the total 568 SAM, 326 children were cured and the remaining 106 and 45 were lost to follow-up and transferred to other care facilities, respectively. Our outcome data compared to the national Sphere reference standards are shown in Table 3.

### Incidence rate of mortality

At the end of follow-up, there were 5,146 days of observation days for an incidence rate of 1.77 deaths per 100 days and the overall probability of death was 19.28% (95% CI: 14.6–40.5).

**Table 1. Baseline socio-demographic characteristics of 568 severe acute malnutrition admitted children in Pawe general hospital from 2015–2019.**

| Variables | Characters | Frequency % |
|---|---|---|
| Sex | Male | 244(42.96) |
| | Female | 324 (57.2) |
| Residence | Urban | 126(22.18) |
| | Rural | 442(77.82) |
| Age | Between 6–24 month | 355(62.68) |
| | Between 24–48 month | 160(28.17) |
| | Above ≥48 month | 53(9.15) |
| SAM types | Wasting (marasmus) | 318(55.99) |
| | Marasmus Kwashiorkor | 153(26.94) |
| | Kwashiorkor (edematous) | 97(17.08) |
| TB | Present | 70 (12.32%) |
| | Absent | 498 (87.68%) |
| HIV | Positive | 40 (7.04%) |
| | Negative | 528 (92.96%) |
| Skin dermatitis | Absent | 353(61.8%) |
| | Present | 217(38.2%) |
| Pneumonia | Present | 296 (52.11%) |
| | Absent | 272 (47.89%) |
| Anemia | Absent | 399(70.75%) |
| | Present | 169(29.75%) |
| Malaria | Present | 42(7.39%) |
| Vomiting | Absent | 281(47.89) |
| | Present | 296(52.13) |
| Fever ≥37.5˚C) | Absent | 265(46.65) |
| | Present | 303(53.35) |
| Diarrhea | Absent | 257(45.2) |
| | Present | 311(54.75) |
| Breastfeeding status | No breastfeed | 207(36.4) |
| | Yes breastfeed | 361(63.56) |
| Admission type | New admission | 457(80.46) |
| | Re-admission | 111(19.54) |
| Nasogastric tube inserted | Yes | 243(41.02) |
| | No | 325(58.98) |
| Vitamin A supplementation | Given | 470 (82.7) |
| | Not given | 98 (17.25) |
| Folic acid | Given | 406 (71.48) |
| | Not given | 162 (28.52) |
| Deworming | Given | 242 (42.61) |
| | Not given | 326 (57.39) |
| Blood transfusion during admission | Given | 123 (21.65) |
| | Not given | 445 (78.35) |

## Predictors of SAM mortality

In the bi-variable analysis living in rural areas, hemoglobin level < 11 g/dl, admission types, nasogastric therapy at admission, vomiting during admission, malnutrition types, diarrhea, MUAC, deworming, WFA, HFA, IV fluid, and breastfeeding status were found to be

**Table 2. Outcomes according to the World Health Organization phases.**

| Indicators | | Phase 1 | Transition phase | Phase 2 | Total |
|---|---|---|---|---|---|
| 1 | Deaths | 32 (35.16%) | 24 (26.37%) | 35 (38.4%) | 91(16.02%) |
| 2 | Cure | 97(29.7%) | 105(32.2%) | 124(38.03%) | 326 (57.39%) |
| 3 | Lost to follow up | 11(10.3%) | 34(32.07%) | 61(57.5%) | 106 (18.66%) |
| 4 | Transferred out | 6(13.3%) | 30(66.6%) | 9(2%) | 45 (7.92%) |
| | Total | 146(25.7%) | 193(33.9%) | 229(40.3%) | 568 (100%) |

**Table 3. Performance indicators achieved at Pawe general and referral hospital from 2015–2019 compared to the national Sphere reference standards (N = 568).**

| Parameter | | Performance of Pawe General hospital (N = ~ 5 years) | SPHERE project reference values | | |
|---|---|---|---|---|---|
| | Indicator of Performance | Achievement | Overall | Acceptable | Alarming |
| 1 | Cure rate | 326 (57.39%) | 77.9% | >75% | <50% |
| 2 | Incidence of death | 91 (16.02%) | 12.3% | <15% | >25% |
| 3 | Lost to follow-up | 106 (18.66%) | 5.2% | <10% | >15% |
| 4 | Transfer out | 45 (7.92%) | 4.4% | ____ | ____ |
| | Total | 568 (100%) | 100 | - - - - - | - - - - - - - |

significant predictors of mortality. In the multivariable Cox-regression analysis, only five variables were found to be predictors of mortality (Table 4). These were: (i) vomiting AHR = 5.1 (1.35–21.1, p = 0.026), (ii) diarrhea AHR = 2.79 (1.46–5.4, p = 0.002), (iii) needing nasogastric therapy AHR = 3.22 (1.65–6.26, p = 0.001) (iv) anemia AHR = 1.89 (1.15–3.2, p = 0.012), and (v) being re-admitted with SAM AHR = 1.7 (1.12–2.8, p = 0.037).

## Discussion

In this retrospective study, we have shown that the overall crude incidence of mortality was 16% with more than half of deaths occurring early in the first and the transition phases. This

**Table 4. Bi-variable and multivariable Cox regression for predictors of mortality.**

| Covariate | Categories | Survival status | | CHR (95%CI) | AHR (95%CI) | P-value |
|---|---|---|---|---|---|---|
| | | Death | Censored | | | |
| Age of children | 6–24 Month | 68(11.9%) | 186(32.74%) | 1 | | |
| | 24–48 month | 22(3.8%) | 145(25.5% | 0.9 (0.57–1.4) | 1.2 (0.73–2.01) | 0.443 |
| | ≥48 month | 1(0.18%) | 46(8.1%) | 0.61 (0.59–4.4) | 2.1 (0.82–3.9) | 0.27 |
| Sex | Male | 30(5.3%) | 214(37.6%) | 1.13 (0.89–1.43) | 0.96 (0.61–1.53) | 0.859 |
| | Female | 61(10.7%) | 263(46.3%) | 1 | 1 | |
| Admission types | New | 67(11.79%) | 183(32.2%) | 1 | 1 | |
| | Re-admission | 24(4.2%) | 294(51.6%) | 1.7 (2.4–6.1) | 1.7 (1.03–2.8) | 0.037* |
| NGT | Yes | 79(13.9%) | 164(28.7%) | 2.3 (2.3–6.99) | 3.22(1.65–6.26) | 0.001* |
| | No | 12(2.2%) | 313(55.09%) | 1 | 1 | |
| Vomiting | Yes | 88(15.5%) | 208(36.5%) | 2.34 (1.28–4.3) | 5.1(1.35–21.2) | 0.026* |
| | No | 3(5.4%) | 269(47.4%) | 1 | 1 | |
| Anemia | Present | 64(11.3%) | 105(18.3%) | 4.2 (2.6–3.67) | 1.89(1.2–3.12) | 0.012* |
| | Absent | 27(4.8%) | 372(65.3%) | 1 | 1 | |
| Diarrhea | Yes | 79(13.9%) | 232 (40.7%) | 5.6 (2.99–10.6) | 2.79 (1.46–5.4) | 0.002* |
| | No | 12(2.1%) | 245(43.01%) | 1 | 1 | |

overall mortality rate is higher compared to other studies in Ethiopia like Mekele hospital, 3.8% [4], Tigray general hospital, 6.65% [4], Dilla referral hospital, 7.57% [11], Bahir-Dare referral hospital, 7.7% [22], and the Hawassa University Comprehensive Specialized Hospital (HU-CSH) in Southern Ethiopia, 10.8% [23]. One difference might be that these are university/tertiary referral hospitals which provide excellent care. A high proportion (a little under 20%) of our patients was lost to follow up which was likely due to their absconding/self-discharging because they could not afford to pay for their care. It is probable that many such children also died so our death rate may be underestimated. We also report that the mean time to death after admission was 13 (±8) days. This is similar to that reported from Gondar specialized referral hospital, 12 days [2], but quicker than the HU-CSH, 17 days [23], Mekele Ayder referral hospital, 41.2 days [4], and Shebedino hospital, 36 days [24].

We identified five independent risk factors for death. Vomiting had the highest risk for death, 5-fold compared to no vomiting on admission, and is consistent with findings at North Gondar [25]. Vomiting may cause or exacerbate dehydration leading to renal impairment and the need for NGT or IV fluids. Moreover, we observed an almost 3-fold death increased with diarrhea in line with the studies in Lusaka, [26], Gondar [25], and in Southern Ethiopia [24]. Diarrhea will also cause hypovolemic and exacerbate the deleterious effects of vomiting. The use of an NGT was also associated with mortality, just over 3-fold, and this association has also been reported in studies from Gondar [2] and Dilla [10] referral hospitals. Our study revealed that anemia was another significant factor for death and was associated with an almost 2-fold increased likelihood of death compared to those with no anemia. This may be explained by the deleterious effects of anemia, a poor response to transfusion, and/or a delayed transfusion due to the large number of patients to attend to. The association of anemia and death has also been reported by others in Gondar [2], Nekemte referral hospital [27], and in South Africa had (2.5-fold risk) [28]. A child may experience more than one episode of SAM, depending on the improvement of the underlying factors during inpatient treatment [29], but if re-admitted with a relapse, then the risk of death is doubled. Other factors that may be important in our setting include serving a poor pastoralist community, delayed presentation, lack of maternal participation in feeding programs, over prescription of intravenous therapies, early discontinuation of treatment due to insufficient financial means [2], and poor adherence of WHO SAM treatment guidelines [11,30]. One study from Uganda also identified WHO defined danger signs on admission, namely, lethargy, hypoglycemia, hypothermia, bradycardia, capillary refill > 2 seconds, weak pulse volume and impaired level of consciousness [12]. We did not record these clinical details nor did we collect data on biochemical indices and the socioeconomic status of caregivers. Therefore, the interpretation and application of our findings for clinical decisions and policy should take into account these limitations.

## Conclusion

The SAM mortality rate was high in our children from one stabilizing center in northern Ethiopia. We identified five important factors for death, vomiting, diarrhea, NGT, anemia, and being readmitted with SAM whilst others have identified important clinical signs associated with death. Health education on seeking early medical care, adherence to guidelines and reducing the early abandonment of treatment because of lack of money may improve child survival in our setting.

## Supporting information

**S1 File. Final data set of SAM.**
(DOCX)

**S2 File. English versions checklists.**
(DTA)

## Acknowledgments

The Pawe Woreda Health Bureau, Pawe, and Assosa hospitals administrative staff members have been supported during the data collection of this research. Our last heartfelt thanks was granted to Mr. Tamirate Shewano of (Assistant Professor of Epidemiology) for his unreserved editing and proofreading of the last version of this manuscript.

## Author Contributions

**Conceptualization:** Fassikaw Kebede, Ayalew Kasaw.

**Data curation:** Fassikaw Kebede, Ayalew Kasaw.

**Formal analysis:** Fassikaw Kebede, Belete Negese, Ayalew Kasaw.

**Funding acquisition:** Fassikaw Kebede, Tsehay Kebede, Ayalew Kasaw.

**Investigation:** Fassikaw Kebede, Atitegeb Abera, Ayalew Kasaw.

**Methodology:** Fassikaw Kebede, Belete Negese, Atitegeb Abera, Getahun Fentaw, Ayalew Kasaw.

**Project administration:** Fassikaw Kebede, Ayalew Kasaw.

**Resources:** Fassikaw Kebede, Tsehay Kebede, Ayalew Kasaw.

**Software:** Fassikaw Kebede, Belete Negese, Getahun Fentaw, Ayalew Kasaw.

**Supervision:** Fassikaw Kebede, Ayalew Kasaw.

**Validation:** Fassikaw Kebede, Tsehay Kebede, Ayalew Kasaw.

**Visualization:** Fassikaw Kebede, Belete Negese, Ayalew Kasaw.

**Writing – original draft:** Fassikaw Kebede, Tsehay Kebede, Ayalew Kasaw.

**Writing – review & editing:** Fassikaw Kebede, Atitegeb Abera, Ayalew Kasaw.

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
