## [Decision Letter · Decision Letter 0]

8 Jun 2021

PONE-D-21-11027

INCIDENCE AND PREDICTORS OF SEVERE ACUTE MALNUTRITION MORTALITY RATE AMONG CHILDRENS FROM 6-59 MONTHS IN STABILIZING CENTER AT PAWE GENERAL HOSPITALS, NORTHWEST, ETHIOPIA 2020.

PLOS ONE

Dear Fassikaw Bizuneh, 

Thank you for submitting your manuscript to PLOS ONE. After careful consideration, we feel that it has merit but does not fully meet PLOS ONE’s publication criteria as it currently stands. Therefore, we invite you to submit a revised version of the manuscript that addresses the points raised during the review process.

Please submit your revised manuscript by the end of June. If you will need more time than this to complete your revisions, please reply to this message or contact the journal office at plosone@plos.org. Please include the following items when submitting your revised manuscript:

We look forward to receiving your revised manuscript.

Kind regards,

Walter R. Taylor

Academic Editor

PLOS ONE

Journal Requirements:

2.Thank you for including your ethics statement: "Ethical clearance was obtained from the ethical review committee of the Debre Markos University, College of Health Sciences (Ref. No: HSC/984/16/12).A formal letter was submitted to Pawe General Hospital for permission to be done entitled to research articles, treatment cure rate and predictors for the cure of severe acute malnutrition in children aged 6-59 months at a stabilizing center in Pawe General Hospital, Northwest Ethiopia.(Since 2015-2019)."

a) Please provide additional details regarding participant consent. In the ethics statement in the Methods and online submission information, please ensure that you have specified (1) whether consent was informed and (2) what type you obtained (for instance, written or verbal, and if verbal, how it was documented and witnessed). If your study included minors, state whether you obtained consent from parents or guardians. If the need for consent was waived by the ethics committee, please include this information.

3.In your Data Availability statement, you have not specified where the minimal data set underlying the results described in your manuscript can be found. PLOS defines a study's minimal data set as the underlying data used to reach the conclusions drawn in the manuscript and any additional data required to replicate the reported study findings in their entirety. All PLOS journals require that the minimal data set be made fully available. For more information about our data policy, please see http://journals.plos.org/plosone/s/data-availability.

4.Thank you for stating the following financial disclosure:

"We authors of this research are an Ethiopian citizens  "

Additional Editor Comments:

The paper needs major revision along the lines of the reviewers. Please do try to find a native English speaker to go over the language.

Reviewers' comments:

Reviewer's Responses to Questions

**Comments to the Author**

1. Is the manuscript technically sound, and do the data support the conclusions?

Reviewer #1: No

Reviewer #2: No

2. Has the statistical analysis been performed appropriately and rigorously? 

Reviewer #1: No

Reviewer #2: Yes

3. Have the authors made all data underlying the findings in their manuscript fully available?

Reviewer #1: Yes

Reviewer #2: No

4. Is the manuscript presented in an intelligible fashion and written in standard English?

Reviewer #1: No

Reviewer #2: No

5. Review Comments to the Author

Reviewer #1: This is an interesting paper addressing a very important topic in Pediatrics health. However, the manuscript is full of English grammar errors, the methods are not clear while results are unstructured. Find my full comments attached.

Reviewer #2: Thank you very much for the opportunity to review this manuscript on incidence and predictors of mortality in children with SAM.

The introduction focuses on malnutrition broadly as well as SAM specifically, but sometimes it is unclear which of these definitions the authors are referring to. It might be beneficial if the authors mainly focus on SAM and the rationale for examining predictors of mortality in children with SAM. There is also a description of past research looking at predictors of mortality described in the introduction, and there have been other studies done in this area in addition to the one cited, so it would be useful for the authors to explain how their study adds to the literature.

This analysis used data from 2008 and the end of 2012. (However, later on, the inclusion criteria are stated as children admitted between 2015-2019 so it is unclear what years were used.) If the admissions were between 2008 and 2012, what are the potential limitations of using data from several years ago, and why were more recent data not considered? The authors should also discuss changes in treatment protocols over time, as updates to the WHO guidelines were released in 2013 for example. It would also be important to explain the admission criteria for this treatment centre.

Minor comments

In the abstract and the introduction, the authors use the idiom “lion’s share” which may not be familiar to all readers, so perhaps this could be changed to more common terminology and to specify whether this is about the absolute number of deaths or mortality rates in sub-Saharan Africa versus other regions.

It appears there may be a typo in the abstract in which authors say that the mortality rate was observed within 4578 days.

In the abstract, is also unclear what the authors mean by “not take in” F-75/F-100 formula. Do the authors mean feeding difficulties or loss of appetite, or that children were not provided these formulas?

The sample size calculation is unclear, and appears that it may be appropriate for a prospective study rather than a retrospective analysis.

The terms kwashiorkor and marasmus should be replaced by oedematous malnutrition and severe wasting, respectively.

In the results section, the authors describe the mean weights of children with SAM, yet it would be more useful to provide z-scores if possible since it is difficult to interpret weights alone.

6. PLOS authors have the option to publish the peer review history of their article (what does this mean?). If published, this will include your full peer review and any attached files.

Reviewer #1: **Yes: **Dr Moses Ngari

Reviewer #2: No

---

## [Author Response · Author response to Decision Letter 0]

27 Jun 2021

Dear Reviewers thanks what you forward comments and question in our manuscript for publication 

Indeed we all editors , reviewers and authors worked for reduction of infant and children mortality 

 especially we authors conduct this original research on the silent killer crisis issue of sever acute malnutrition among children admitted in stabilizing center . 

All efforts made for benefit of the study subject based on scientific intervention . The benefit of publication of this research is multi lateral for Ethiopia, also in study area .There for Please consider in the positive way for publication process .

---

## [Editor Report · Decision Letter 1]

7 Jul 2021

PONE-D-21-11027R1

INCIDENCE AND PREDICTORS OF SEVERE ACUTE MALNUTRITION MORTALITY RATE CHILDREN AGE 6-59 MONTHS ADMITTED IN STABILIZING CENTER PAWE GENERAL HOSPITAL, NORTHWEST ETHIOPIA 2020

PLOS ONE

Dear Dr. Kebede, 

Thank you for submitting your manuscript to PLOS ONE. After careful consideration, we feel that it has merit but does not fully meet PLOS ONE’s publication criteria as it currently stands. Therefore, we invite you to submit a revised version of the manuscript that addresses the points raised during the review process.

We look forward to receiving your revised manuscript.

Kind regards,

Walter R. Taylor

Academic Editor

PLOS ONE

Dear Fassikaw Bedebe,

I have been though the paper in some detail and seen that you have made a lot of effort to improve it.

The quality of the English remains quite low and must be improved before I can accept the paper. Please, if you can, find a native speaker.

A few general comments. The paper is quite long and in repetitive in places. Please use line numbers for the next revision so I can more easily point out any changes.

Here are some additional points.

Abstract

Replace lion’s share with most

paw – is Pawe hospital, please correct

Anaemia is defined as Hb < 11 g/dL in the text but it is < 10 g/dL in the Abstract. Please correct.

Study design & area

Not clear about the percentages of beds – 29% and 25% are stated.

Independent variables

The terms kwashiorkor and marasmus should be replaced by oedematous malnutrition and severe wasting, respectively.

Define all abbreviations like ARTI WFA etc

MUAC definitions vary from < 115cm to <=115 cm. Please correct this.

The inclusion and exclusion criteria are the same

Dependent variable. It is not clear when death has to take place – is it as an inpatient in the stabilising centre or some time after discharge?

CO morbidity

One in 5 = 20% not 11.98%. Please just report the %s and numerators and denominators

Remove and etc

Antimalarial treatment i.e. short course or was it really antimalarial prophylaxis ? Please state which drugs were used and for how long.

Please tell us what is in F75and F100 and what the indications are to give one or the other

Mortality incidcne

I do not understand where the median follow up period of 13.5 days comes from. Please tell us how long children were admitted in the SC and how long the follow up period was after discharge.

IV is a parenteral method so please delete.

References

The text has a reference 29 but the reference list stops at 20.

Repetitive text:

Descriptive analyses such as tables, graphs, Kaplan–Meier survival curves, and the log-rank test were performed.

Discussion

The mean waiting time is reported but this is not defined in the Methods section. Please clarify.

I do not see the relevance of mentioning stunting and overweight.

Table 3.

Please explain I the text about different treatment phases.

Please send back the paper when it is ready.

yours sincerely,

Walter Taylor

---

## [Author Response · Author response to Decision Letter 1]

15 Aug 2021

Thank for devoting your invaluable time with my manuscript reviewing 

 In fact Sever acute malnutrition is silent killing crisis in Ethiopia especially on under-five children

 so please consider

---

## [Editor Report · Decision Letter 2]

31 Aug 2021

PONE-D-21-11027R2

INCIDENCE AND PREDICTORS OF SEVERE ACUTE MALNUTRITION MORTALITY RATE FOR CHILDREN AGE 6-59 MONTHS ADMITTED IN STABILIZING CENTER, NORTHWEST ETHIOPIA 2020

PLOS ONE

Dear Mr. Kedebe , 

Thank you for submitting your manuscript to PLOS ONE. After careful consideration, we feel that it has merit but does not fully meet PLOS ONE’s publication criteria as it currently stands. Therefore, we invite you to submit a revised version of the manuscript that addresses the points raised during the review process.

We look forward to receiving your revised manuscript.

Kind regards,

Walter RJ Taylor

Academic Editor

PLOS ONE

Additional Editor Comments:

Dear Mr. Kedebe,

I have read the paper and it is an improvement on the previous revision. However, the English is not yet at a standard that can be accepted. Please find a good or native speaker to revise it.

There are also a number of areas in the paper that are unclear and inconsistent.

These are detailed below.

Yours sincerely,

Bob Taylor.

The death rate percentages vary within the manuscript. Please sort this out.

The Abstract talks of a a “means follow….” But later refers to this as “waiting time.” Both are unclear to me, Please clarify.

The Abstract reports a survival of 88.23% when the overall mortality is 16.7%. The total exceeds 100%. Please clarify.

Page 3

However, 50% of children with SAM die during treatment…… reference 13 reports a cumulative survival rate of ~85% so where does the 50% mortality come from?

and in fact, the majority (1.5%–40%) – In English, the majority usually means > 50%. Delete this.

Page 10

Nearly one in five (11.98%), and one fourth (29.2%) study participants

1 in 5 = 20%, a lot higher than 11.98%. Please amend. Please always show the numerators when you show a %.

Page 11. Please delete etc.

Page 11. 91 (16.23%) then IDR = 16.29%. Please note the death rate in the Abstract 16.7% in the Abstract. Please make sure the crude death rate is the same throughout the paper.

I don’t see the point in reporting the death rate after 1, 2 and 3 weeks etc. We should be able to see this for the KM curve.

Page 12

Among 568, study participants, 91(16.29%) SAM admitted cases developed the event of inters. INTEREST

Page 13

Death rate 16.73% and the 16.7 on the next line.

The average (mean) waiting time of 13.5 days in this study is in agreement with the finding in Gondar 12days[5].

Is this correct? What is the waiting time? It has not been defined in the M&M section.

I do not understand what this sentence means:

“The elucidation for this finding might be an instant similarity management team with a medical supply set.”

Page 14

“Furthermore, as indicated by the results of this study, the cumulative survival probabilities on 7th, 14th, 21st, and 28th day was 97.8%,84.48%,60.5%,56.5%and 48.9%, respectively.”

Consider removing this. What does it tell us? Moreover, we have 4 time intervals but five death rates.

Page 15

Pease complete this sentence: This is comparable with a study conducted by work et al. [37],

Table 1.

What drug is this – Premaquinin?

Table 2. Mean waiting time - what is this?

Table 3 Death rate 16.02%. please correct.

Figure 3. This shows that the cumulative survival rate over 30 days is 0%. Please correct.
---

## [Author Response · Author response to Decision Letter 2]

4 Sep 2021

Thanks dear reviewers for devoting your time and enthusiastic way of revising my manuscript for fruit full ways of my efforts

---

## [Editor Report · Decision Letter 3]

16 Sep 2021

PONE-D-21-11027R3I INCIDENCE AND PREDICTORS OF SEVERE ACUTE MALNUTRITION MORTALITY RATE AMONG CHILDREN AGED 6-59 MONTHS ADMITTED AT PAWE GENERAL HOSPITAL STABILIZATION CENTER, NORTHWEST ETHIOPIA 2021PLOS ONE

Dear Mr. Kebede, 

Thank you for submitting your manuscript to PLOS ONE. After careful consideration, we feel that it has merit but does not fully meet PLOS ONE’s publication criteria as it currently stands. Therefore, we invite you to submit a revised version of the manuscript that addresses the points raised during the review process.

We look forward to receiving your revised manuscript.

Kind regards,

Walter RJ Taylor

Academic Editor

PLOS ONE

Journal Requirements:

Additional Editor Comments (if provided):

Dear Mr. Kebede,

Thankyou for sending an updated revision.

The quality of the English is better but still needs to be improved before the paper can be published. Most of the errors are easily fixed by a native speaker.

I have a few specific points.

Abstract

“During SAM inpatient treatment recovery, transfer out, lost follow-up, and death was found 326 (57.39%), 106(18.66%), 46 (8.10%), and 91(16.02%), respectively. The overall incidence of the mortality rate was 16.03 per 100 (95%CI: 13.86; 20.04) person-days observations.”

I do not understand why there are two different rates for mortality.

I would suggest you replace “Operational words” with Definitions.

“or mid-upper-arm circumference ≤115 mm, or presence of bilateral edema, and failed appetite test should be admitted for inpatient care[23].”

The current MUAC definition is <115 mm. Please change. The cited reference is fine.

Discussion

The Sphere cut off is <10% but <11% in the Abstract. Please correct.

Conclusion

A majority usually means > 50% so please amend this sentence. Why not say that mortality in your series was high and half the deaths occurred within 2 weeks.

Then list the key factors and suggest clinical care be focused preferentially on these risk factors.

Tables.

Table 1.

I cannot find the drug “Permaquinin” on Google or in the Ethiopian Guidelines for treating SAM. Please tell us the name of this drug.

Table 2.

Please explain what “Waiting time” means.

Table 4.

This is unclear. Is it showing an overall survival rate for all the children? If so, 21.09% survival is much less than the reported survival in the paper:

“At the end of this study period, 326 (57.39%) admitted under-five children had been cured”

Figures.

Labelling of the Y axes of the 3 figures has been cut away. Please make sure we can see the labels.

Figure 3.

This makes no sense to me. It shows a very low survival rate. I suggest you either remove this Figure or put in the correct KM curve.

Figure 7.

Analysis is not spelt correctly on the X-axis.

---

## [Author Response · Author response to Decision Letter 3]

17 Sep 2021

Thanks to be with my manuscript and devoting your time of reviewing

---

## [Editor Report · Decision Letter 4]

5 Oct 2021

PONE-D-21-11027R4INCIDENCE AND PREDICTORS OF SEVERE ACUTE MALNUTRITION MORTALITY RATE AMONG CHILDREN AGED 6-59 MONTHS ADMITTED AT PAWE GENERAL HOSPITAL, NORTHWEST ETHIOPIA 2021PLOS ONE

Dear Mr. Kebede, 

Thank you for submitting your manuscript to PLOS ONE. After careful consideration, we feel that it has merit but does not fully meet PLOS ONE’s publication criteria as it currently stands. Therefore, we invite you to submit a revised version of the manuscript that addresses the points raised during the review process.

We look forward to receiving your revised manuscript.

Kind regards,

Walter RJ Taylor

Academic Editor

PLOS ONE

Additional Editor Comments:

Dear Mr. Kebede,

Thankyou for sending an updated revision.

The quality of the English is a little better but it is clear that you have not been able to find a native speaker to help you with the language.

Below are several of the more serious errors in language that I request you amend.

Introduction

Childhood under-nutrition incorporates a combination of nutrition disorders that included underweight

“that include…”

recent failure to received

“to receive”

Globally in 2018, 52 Million under-five-years -children in one in twelve of this age group -are

suff ering from SAM

“Globally in 2018, 1 in 12 of the estimated 52 million children under five had SAM.”

The one underline reason is health care provider's poor adherence to world health organization SAM therapeutic guidelines [1], besides to late coming of the patient for treatment[11]

“One important reason is poor adherence to… and another is late presentation.”

Malnutrition in Ethiopia is a long-term silent killing crisis, especially for infants, and children yet contribute to an estimated 270,000 deaths of under-five children each year [12].

“…, especially for infants, and contributes to an……”

Despite Ethiopia launching the Seqota declaration to end up severe acute malnutrition death by

“to end…”

Although, the magnitude of mortality among SAM admitted under-five children were a good document in Ethiopia, survival

“Although the….. children is well documented in….”

Methods

The Pediatric ward is among the five inpatient departments found in Pawe general hospital with 252 beds for inpatient treatment [20, 21]. Thought, the stabilizing center is separately built with single-based beds for children in SAM patient's treatments.

“The Pediatric ward is among the five inpatient departments found in Pawe general hospital with 252 beds for inpatient treatment [20, 21] and a separate stabilizing center for children with SAM.”

In all phases admitted children were, treated and prevented for infections, resuscitated, restored electrolyte balance, hypoglycemia, and hypothermia.

“In all phases admitted children are treated empirically for infections, hypoglycemia and hypothermia and resuscitated to restore electrolyte balance.”

Results

In this study, the outcome of interest is the SAM inpatient death, and death was defined as

following after inpatient admission to report of death during treatment observation.

“In this study, the outcome of interest was inpatient death due to SAM.”

Operational words

Hgb levels mainly classified into two ways: No anemia >11 g/dL, and Anemia≤10.9 g/dl [25].

Please check this. To me it should be <11 and >=11 g/dL.

Result

and the mean age was participants was 26.28 (SD = ±16.04) months.

“and the mean age was 26.28 (SD = ±16.04) months.”

The majority, 356 (62.68 %) of children have aged between 6-24 months years Vs. bottom number 47 (8.27%)of ≥48-month children.

This sentence is very unclear. Suggest you change it to: “ The majority of children, 356 (62.68 %), were aged between 6-24 months.”

majority of 38(41.11%) deaths were reported at Phase I within 24-144 hours after admission. Inversely, 124(38.6%) SAM admitted children declared as cured at phase II (Table3).

“majority of deaths, 38 (41.11%), were reported during Phase I within 24-144 hours after admission whilst 124 (38.6%) of SAM children were cured in phase II (Table3). “

Which was higher than the national SPHERE reference (<11%) [9]. This difference might be due to delayed presentation to the health institution (SC) and discontinuation of treatment by the financial limit to buy accessory drugs and foods after admissions [5] beside to worthless contribution of health care providers poor adherence to world health organization's SAM treatment guidelines [1, 27].

“This was higher than the national SPHERE reference (<11%) [9] and might be due to delayed presentation to the health institution (SC), early discontinuation of treatment due to insufficient financial means admissions [5], and poor adherence to WHO SAM guidelines [1, 27].”

A child may experience more than one episode of SAM relapsing, depending on the improvement of the underlying factors during inpatient [31]. Likewise, in this report death hazard was two times increased for re-admitted (relapsed) SAM cases as compared with new SAM inpatient SAM cases.

“A child may experience more than one episode of SAM, depending on the improvement of the underlying factors during inpatient treatment [31]. SAM relapses have a two-fold hazard of death.”

A new analysis of the multiple burdens of malnutrition within nations provides novel insights into the degree of childhood stunting, anemia, and overweight. This exposes 29% of the experience of re-admissions[32].

This sentence does not make any sense. I suggest you delete it.

The finding in Lusaka, Zambia [33], Gonder referral hospitals [34], Southern Ethiopia. [2], and Irena [33] revealed that SAM children admitted with baseline diarrhea were increased the risk of mortality rate as compared with counter groups, which is similar with twice hazards of death increased in our study finding. This is might be due to children having more than three episodes of stool loss per day is highly associated with the cleaning of micronutrients from their body and shunting of intracellular fluids causes hypovolemic shock.

“In Lusaka, Zambia [33], Gonder referral hospitals [34], Southern Ethiopia. [2], and Irena [33], SAM children admitted with baseline diarrhea had an increased risk of death, consistent with our two-fold increase in death, which may be due to combination of fluid and electrolyte loss and possible hypovolemic shock.”

Delete this sentence: This could be due to the shrinking of the intracellular potassium pumping balance of the body homeostatic.

Limitations

it failed to consider a broad range of factors like some biochemical and individual bases economic status of caregivers’ might introduce a high array of missing confounders. As such, the interpretation and application of the finding for decision and policy direction should account for these inherent limitations of the study.

“it failed to consider a broad range of factors like some biochemical indices and the socioeconomic status of care givers. Therefore, interpretation the interpretation and application of the finding for decision and policy direction should account for these inherent limitations of the study.”

Figure 1.

Please change the title to something like “Number of SAM children admitted to Pawe hospital.”

Figure 2. Kaplan-Meier estimated survival……

---

## [Author Response · Author response to Decision Letter 4]

7 Oct 2021

Dear editor we would like to thanks to be with this manuscript

---

## [Editor Report · Decision Letter 5]

25 Oct 2021

PONE-D-21-11027R5INCIDENCE AND PREDICTORS OF SEVERE ACUTE MALNUTRITION MORTALITY RATE AMONG CHILDREN AGED 6-59 MONTHS ADMITTED AT PAWE GENERAL HOSPITAL, NORTHWEST ETHIOPIA 2021PLOS ONE

Dear Dr. Kebede, 

Thank you for submitting your manuscript to PLOS ONE. After careful consideration, we feel that it has merit but does not fully meet PLOS ONE’s publication criteria as it currently stands. Therefore, we invite you to submit a revised version of the manuscript that addresses the points raised during the review process. The standard of English needs to be improved and I have been through the paper and made some suggestions.

I do not understand the national protocol and the 11% - is <11% the target death rate ?

Hopefully, after you include the suggestions, the English will be acceptable.

I do not see the value of Table 4 and the multiple Figures when Table 5 contains the key information on the independence of the variables. Moreover, there are issues with the Figures.

Figure 1 is inconsistent with the reported death rate in the text of 16.03%. From the Figure 1 the cumulative risk of death is about 75%. I suggest you remove Figure 1.

The other Figures are not of high quality, are difficult to read because the legend obscures the graphs, and some contain spelling mistakes. When estimating the hazard ratios from the graphs, they appear quite different to those in Table 5. In Figure 5, was the cumulative death rate 75% in new cases and that in readmitted cases 100%? From Table 5, the estimated death rate in the new cases was 26.8% (67/250) and 7.5% (24/318) in the readmitted cases. Your data are inconsistent.

I suggest you recheck all the data in Tables 4 and 5 and the delete all the Figures. Show the data in Table 4 so we can e.g. see the death rates in each category and the p values. There is no need to show the chi squared values as well.

Please submit your revised manuscript by 15th of November. If you will need more time than this to complete your revisions, please reply to this message or contact the journal office at plosone@plos.org. Please include the following items when submitting your revised manuscript:A rebuttal letter that responds to each point raised by the academic editor and reviewer(s). You should upload this letter as a separate file labeled 'Response to Reviewers'.A marked-up copy of your manuscript that highlights changes made to the original version. You should upload this as a separate file labeled 'Revised Manuscript with Track Changes'.An unmarked version of your revised paper without tracked changes. You should upload this as a separate file labeled 'Manuscript'.

We look forward to receiving your revised manuscript.

Kind regards,

Walter RJ Taylor

Academic Editor

PLOS ONE

---

## [Author Response · Author response to Decision Letter 5]

26 Oct 2021

Thanks for devoting your time to review this manuscript

---

## [Editor Report · Decision Letter 6]

10 Nov 2021

PONE-D-21-11027R6INCIDENCE AND PREDICTORS OF SEVERE ACUTE MALNUTRITION MORTALITY RATE AMONG CHILDREN AGED 6-59 MONTHS ADMITTED AT PAWE GENERAL HOSPITAL, NORTHWEST ETHIOPIA 2021PLOS ONE

Dear Mr. Kebede, 

Thank you for submitting your manuscript to PLOS ONE. After careful consideration, we feel that it has merit but does not fully meet PLOS ONE’s publication criteria as it currently stands. Therefore, we invite you to submit a revised version of the manuscript that addresses the points raised during the review process.

The English still needs to be improved so I would suggest you send me the latest version of word file so I can assist in improving the text.

Please submit the word file by the Dec 25 2021 11:59PM. 

We look forward to receiving your word file. 

Kind regards,

Walter RJ Taylor

Academic Editor

PLOS ONE
---

## [Author Response · Author response to Decision Letter 6]

14 Nov 2021

I had address the issue raised the language part by senior researcher and for final step it is resolved please make it accept and make swift for finale publications steps thanks for your comment

---

## [Editor Report · Decision Letter 7]

7 Jan 2022

PONE-D-21-11027R7INCIDENCE AND PREDICTORS OF SEVERE ACUTE MALNUTRITION MORTALITY RATE AMONG CHILDREN AGED 6-59 MONTHS ADMITTED AT PAWE GENERAL HOSPITAL, NORTHWEST ETHIOPIA 2021PLOS ONE

Dear Mr. Kebede, 

Thank you for submitting your manuscript to PLOS ONE. After careful consideration, it requires minor amendment.Therefore, we invite you to submit a revised version of the manuscript. Please submit your revised manuscript by  Feb 21, 2022. If you will need more time than this to complete your revisions, please reply to this message or contact the journal office at plosone@plos.org. Please include the following items when submitting your revised manuscript:A rebuttal letter that responds to each point raised by the academic editor and reviewer(s). You should upload this letter as a separate file labeled 'Response to Reviewers'.A marked-up copy of your manuscript that highlights changes made to the original version. You should upload this as a separate file labeled 'Revised Manuscript with Track Changes'.An unmarked version of your revised paper without tracked changes. You should upload this as a separate file labeled 'Manuscript'.If applicable, we recommend that you deposit your laboratory protocols in protocols.io to enhance the reproducibility of your results. Protocols.io assigns your protocol its own identifier (DOI) so that it can be cited independently in the future. For instructions see: https://journals.plos.org/plosone/s/submission-guidelines#loc-laboratory-protocols. Additionally, PLOS ONE offers an option for publishing peer-reviewed Lab Protocol articles, which describe protocols hosted on protocols.io. Read more information on sharing protocols at https://plos.org/protocols?utm_medium=editorial-email&utm_source=authorletters&utm_campaign=protocols.

We look forward to receiving your revised manuscript.

Kind regards,

Walter RJ Taylor

Academic Editor

PLOS ONE
---

## [Author Response · Author response to Decision Letter 7]

11 Jan 2022

Thanks for your comment 

For the purpose of good information containing on back ground information of the study area, we were changed reference No.19 of the old version articles entitled as “UNICEF. Situation Analysis of Children and Women:Beni shangul-Gumuz Region. REPORT 2019” by 

“Time to Develop and Predictors for Incidence of Tuberculosis among Children Receiving Antiretroviral Therapy; Tuberculosis Research and Treatment 2021, 2021:6686019.” Kebede F, Kebede T, Kebede B, Abate A, Jara D, Negese B, Shawano T.

However by the request of acadmic editore ways of new papers cited or retracted have been forbiden as a rules of PLOS ONE Journal Requirements

 We have been corrected to be the first article on citation at reference No.19”

 In short we have maintiend previouse citated article on the paper emetoled as “UNICEF. Situation Analysis of Children and Women:Beni shangul-Gumuz Region. REPORT 2019”

Please look at reference No-19

---

## [Editor Report · Decision Letter 8]

17 Jan 2022

INCIDENCE AND PREDICTORS OF SEVERE ACUTE MALNUTRITION MORTALITY IN CHILDREN AGED 6-59 MONTHS ADMITTED AT PAWE GENERAL HOSPITAL, NORTHWEST ETHIOPIA

PONE-D-21-11027R8

Dear Dr. Kebede,

We’re pleased to inform you that your manuscript has been judged scientifically suitable for publication and will be formally accepted for publication once it meets all outstanding technical requirements.

Kind regards,

Walter RJ Taylor

Academic Editor

PLOS ONE
---

## [Editor Report · Acceptance letter]

17 Feb 2022

PONE-D-21-11027R8 

Incidence and predictors of severe acute malnutrition mortality in children aged 6-59 months admitted at Pawe general hospital, Northwest Ethiopia 

Dear Dr. Kebede:

I'm pleased to inform you that your manuscript has been deemed suitable for publication in PLOS ONE. Congratulations! Your manuscript is now with our production department. 

Kind regards, 

on behalf of

Dr. Walter RJ Taylor 

Academic Editor

PLOS ONE